# Assessing genotypes of buffel grass (*Cenchrus ciliaris*) as an alternative to maize silage for sheep nutrition

Sultan Singh[1], Pushpendra Koli[1,2]*, Tejveer Singh[1], Madan Mohan Das[1], Subhendu Bikash Maity[1], Krishna Kunwar Singh[1], Rohit Katiyar[1], Asim Kumar Misra[1], Sanat Kumar Mahanta[1], Manoj Kumar Srivastava[1], Uchenna Young Anele[3], Olatunde Akeem Oderinwale[3], YongLin Ren[2]*

1 ICAR-Indian Grassland and Fodder Research Institute, Jhansi, UP, India, 2 College of Environmental and Life Sciences, Murdoch University, Murdoch, WA, Australia, 3 North Carolina Agricultural and Technical State University, Greensboro, NC, United States of America

* kolipushpendra@gmail.com (PK); y.ren@murdoch.edu.au (YR)

**Data Availability Statement:** All relevant data are within the manuscript and its Supporting Information files. Minimal data set is uploaded as supporting information.

## Abstract

Nutritive value of five *Cenchrus ciliaris* (buffel grass) genotypes (IG96-50, IG96-96, IG96-358, IG96-401 and IG96-403) weredetermined. Their sugar contents (>70 mg/g of dry matter) and ensiling potential were evaluated using *in vitro* batch culture and *in vivo* studies. Research indicated significant differences ($P < 0.05$) in the dry matter, organic matter, ether extract, neutral detergent fiber, acid detergent fiber, cellulose and lignin contents of the *C. ciliaris* genotypes tested. Genotypes also differed ($P < 0.05$) in total carbohydrates, structural carbohydrates, non-structural carbohydrates and protein fractions. Genotype IG96-96 had the lowest total digestible nutrients, digestible energy and metabolizable energy contents (377.2 g/kg, 6.95 and 5.71 MJ/kg of dry matter, respectively), and net energy values for lactation, maintenance and growth. After 45 days of ensiling, *C. ciliaris* silages differed ($P < 0.05$) in dry matter, pH, and lactic acid contents, and their values ranged between 255–339, 4.06–5.17 g/kg of dry matter and 10.8–28.0 g/kg of dry matter, respectively. Maize silage had higher ($P < 0.05$) Organic Matter (919.5g/kg of dry matter), ether extract (20.4g/kg of dry matter) and hemi-cellulose (272.3 g/kg of dry matter) than IG96-401 and IG96-96 silages. The total carbohydrates and non-structural carbohydrates of maize silage were higher ($P < 0.05$), while structural carbohydrates were comparable ($P < 0.05$) with *C. ciliaris* silages. Sheep on maize silage had ($P < 0.05$) higher metabolizable energy, lower crude protein, and digestible crude protein intake (g/kg of dry matter) than those on *C. ciliaris* silage diets. Nitrogen intake and urinary-N excretion were higher ($P < 0.05$) on genotype IG96-96 silage diet. Overall, this study suggested that certain *C. ciliaris* genotypes, notably IG96-401 and IG96-96, exhibited nutritive values comparable to maize silage in sheep studies, offering a promising avenue for future exploration as potential alternatives in diversified and sustainable livestock nutrition programs.

**Funding:** The author(s) received no specific funding for this work.

**Competing interests:** The authors have declared that no competing interests exist.

**Abbreviations:** C, *ciliaris*: *Cenchrus ciliaris*; DDM, digestible dry matter; DMI, dry matter intake; tCHO, : total carbohydrate; $C_A$, rapidly degradable carbohydrates (CHOs); $C_{B1}$, intermediately degradable starch and pectin; $C_{B2}$, slowly degradable cell wall; $C_C$, unavailable/lignin-bound cell wall; $P_A$, non-protein nitrogen; $P_{B1}$, buffer-soluble protein; $P_{B2}$, neutral detergent-soluble protein; $P_{B3}$, acid detergent-soluble protein; $P_C$, indigestible protein, N-nitrogen; $H_2SO_4$, hydrogen sulfuric acid; $NE_L$, Net energy values for lactation; $NE_G$, Net energy values for growth; $NE_M$, Net energy value for maintenance; $NH_3$-N, ammonia nitrogen; RFV, relative feed value; NFC, Non-fiber carbohydrate; ME, metabolizable energy; CPCSEA, committee for the purpose of control and supervision of experiments on animals.

## Introduction

India possesses a substantial livestock population, totaling 536 million [1]. However, individual productivity per head, encompassing aspects such as milk, meat, wool, etc., remains low, primarily attributed to insufficient nutrition stemming from inadequate feed and fodders, coupled with a diminished genetic potential. The gap between demand and supply for green fodder is more than the demand-supply of dry roughages [2]. India has a large grazing area which includes 69.4 million hectares of forest, 10.9 million hectares of permanent pasture/grazing lands, 13.7 million hectares of cultivable wasteland and 54.0 million hectares of area under common property resources from where surplus grass biomass is available after their active growth during monsoon [2]. These surplus monsoon grasses harvested at early bloom stage can be conserved as silage to provide moderate quality fodder for feeding, especially during dry periods. Silage is an important component of many livestock production systems and it was determined that the effectiveness of producing and using grass is crucial for livestock farming in tropical areas [3]. Tropical range grasses are often fed green or as hay, but rarely as silage mainly because of their low dry matter, less water-soluble carbohydrate, higher buffering capacity and low energy contents [4] which restrict fermentation. With the adoption of tropical grasses silage technology, tropical range grasses have been successfully ensiled with additives [5–8]. *Cenchrus ciliaris* (buffel grass) holds significance as a tropical pasture grass naturally found in Northern Africa, the Middle East, and throughout India. It is widely employed as a forage grass in India [9]. It is drought tolerant and if well fertilized, *C. ciliaris* herbage yield may be up to 24 t dry matter per ha [10]. At the early flowering stage, hay prepared from the grass is of medium quality and rarely used for silage because of its low sugar and moisture contents in the semi-arid regions. Efforts have been made to breed *C. ciliaris* cultivars for improved nutritive value and higher fodder yield particularly in Australia [11]. In India, while a few varieties have been developed for higher biomass [12], there hasn't been any research effort aimed at breeding varieties capable of ensiling without the need for silage additives. This lack of research in developing such varieties hinders the adoption of grass silage technology, particularly on small to medium-sized farms. To address such research gaps, we undertook a nutritional evaluation of *C. ciliaris* germplasm for sugar content (>70 mg/g of dry matter) and ensiling properties. From this evaluation, five genotypes, namely IG-96-50, IG96-96, IG96-358, IG96-401 and IG96-403 were selected and assessed for their nutritive value and compared with maize silage in a digestibility study. The objectives of this study were to evaluate ensiling potential of selected *C. ciliaris* genotypes with respect to their carbohydrates and protein fractions and their nutritive value, and how these genotypes compared to maize silage in terms of nutritional composition when fed to sheep.

## Materials and methods

### Experiment 1: Screening assays for different genotypes *Cenchrus ciliaris*

**Fresh raw materials.** Samples were collected randomly at the flower initiation stage from three locations in each plot and pooled into three representative samples for each genotype. Harvested samples were dried at 100°C for 72 h for dry matter estimation and at 60°C for 72 h for biochemical estimations [13]. Dried samples were ground through a 1-mm sieve using a Willey mill and then stored at 4°C for further nutritional and *in vitro* analyses.

**Experimental location and establishment.** The experiment was conducted at the Indian Grassland and Fodder Research Institute, Jhansi, India. This location receives an annual precipitation of 906.5 mm, with an average annual temperature of 26°C and it is located at an altitude of 271 m. The soil prevalent in the area is classified as inceptisol, featuring surface

textures of loamy and clay loam. *C. ciliaris* genotypes IG96-50, IG96-96, IG96-358, IG96-401and IG96-403 with more than 70 mg/g of dry matter sugar contents were selected from a two-year multidisciplinary nutritional evaluation of 79 genotypes aimed to improve nutritional quality and yield of *C. ciliaris*. The genotypes were planted in plots measuring 15m × 30m, with two plots allocated for each genotype. The spacing between lines and plants was $50 \times 50$ cm during the Kharif season (July to October) in 2015. Prior to sowing, a basal fertilizer of 42 kg N/ha and 42 kg $P_2O_5$/ha was applied, and a topdressing of 46 kg N/ha was administered one month after germination during the floral initiation stage.

**Silage preparation.** For laboratory ensiling, the five *C. ciliaris* genotypes were harvested and wilted for 2h. The samples were chopped 1.5 cm pieces using a manually operated chaffing machine and then placed into plastic containers measuring 25.5 cm in length, 13 cm in diameter, with a volume of 5 kg for each genotype. This was done in triplicate. The chaffed samples were filled and pressed manually with a broad-based wooden rod to exclude as much air as possible and then containers were capped and sealed with adhesive tape to prevent air entering. After 45 days, silage containers were opened, and representative samples were analyzed for silage chemical composition and fermentation quality.

**Fermentation profiles and chemical analysis and fermentation profiles.** *Chemical composition*. The dry matter, nitrogen, ether extract and ash contents of green (fresh), silage, and concentrate mixture samples were estimated as described [14]. Nitrogen values were multiplied by 6.25 to convert to crude protein values. Neutral detergent fiber, acid detergent fiber, cellulose and lignin (sa) were estimated by sequential procedure [15] using a fiber analyzer (Fibra Plus FES 6, Pelican, Chennai, India). Both neutral detergent fiber and acid detergent fiber were expressed inclusive of residual ash. Lignin (sa) was determined by solubilization of cellulose with sulfuric acid in the acid detergent fiber residue [15]. Cellulose was calculated as the difference between acid detergent fiber and lignin(sa) in the sequential analysis. Hemicellulose was calculated as the difference between neutral detergent fiber and acid detergent fiber.

*Estimation of sugar contents*, *carbohydrate and protein fractions*. Sugar content was determined using approximately 100 mg of ground samples treated with 10 ml of 80% ethanol at 80˚C for 30 min, followed by centrifugation at 10000 rpm for 10 min. The dilution was made as previously reported [16]. The sugar was calculated using the anthrone method, with the developed blue color measured using a UV-spectrophotometer (LabIndia3000) at 630 nm [17].

The determination of the total carbohydrate (g/kg of dry matter) of green (fresh) and silage samples involved subtracting the crude protein, ether extract, and ash contents from 1000. It is broadly divided into 4 fractions including, rapidly degradable carbohydrates including sugars ($C_A$), intermediately degradable starch and pectin ($C_{B1}$), slowly degradable cell wall ($C_{B2}$), unavailable/lignin-bound cell wall ($C_C$). Structural carbohydrate was calculated as the difference between neutral detergent fiber and neutral detergent insoluble protein, and non-structural carbohydrates were estimated as the difference between total carbohydrates and structural carbohydrates [18]. The crude protein fractions of green and silage samples were partitioned into five fractions according to the Cornell Net Carbohydrate and Protein System [19] with modifications [20]. Non-protein N ($P_A$) were estimated as the difference between total crude protein and true crude protein precipitated with sodium tungstate (0.30 M) and 0.5 M sulphuric acid; buffer soluble protein ($P_{B1}$), calculated as the difference between true protein and buffer-insoluble protein estimated with borate-phosphate buffer (pH 6.7–6.8) and freshly prepared 0.10 sodium azide solution. Neutral detergent soluble protein ($P_{B2}$) was estimated as buffer-insoluble protein minus neutral detergent insoluble protein, whereas acid detergent soluble crude protein ($P_{B3}$), was estimated as the difference between neutral detergent insoluble

protein and acid detergent insoluble crude protein. Acid detergent insoluble protein ($P_C$), which is unavailable is determined in acid detergent fiber residue.

*Silage analysis*. To prepare the silage extract, 100g of fresh silage samples were subjected to drying in a hot air oven at 60°C until reaching a constant weight. The dry matter content was subsequently adjusted using a specific equation [21]. For the measurement of pH and lactic acid, a 20g portion of fresh silage sample was placed in a beaker, followed by the addition of 100 ml of tepid water. This mixture was then positioned in a water bath equipped with a shaker, set at 100 rpm and maintained at 30°C for a duration of 30 minutes. After agitation, the contents were filtered through a filter paper, and the resulting filtrate was homogenized to obtain the silage extract, ready for further analysis and evaluation.

A portion of filtrate was measured using a digital pH meter (Systronic 360), while the remaining filtrate was utilized for lactic acid estimation [22]. Lactic acid determination involved mixing 1 ml of the extract with 0.05 ml of 4% CuSO4 solution, addition of 6 ml of $H_2SO_4$ dropwise, boiling for 5 min, cooling, adding 0.1 ml p-hydroxyphenyl reagent, incubating at 30°C for 30 min, and measuring the developed blue color at 560 nm using a UV-spectrophotometer (LabIndia 3000) [9]. The estimation of ammonia nitrogen in silage extracts was estimated as described previously [23].

## Experiment 2: *In vivo* digestibility by sheep

**Sheep feeding.**   For the sheep feeding study, the entire biomass from two plots of genotypes (IG96-96 and IG96-401) was harvested and chaffed with an electrically operated chaff cutter. The chaffed material was then placed into separate pits measuring approximately 1.3 m x 2.0 m, having earthen base and cemented side walls. During the filling process, chaffed material was manually compressed by 2–3 people walking over the material to expel as much air as possible. To create anaerobic conditions in the pits, polythene sheets covered them, and thatching materials and sand placed on top. After 45 d, the pits were opened and the silage used to feed the sheep, utilizing maize silage as a control diet.

**Sheep *in vivo* digestibility.**   Fifteen adult rams of *Jalauni* breed were randomly assigned to three silage groups (made from genotypes IG96-401, IG96-96 and maize), Each treatment comprised five animals and subjected to all three silage groups with mean body weights of 21.7 ±1.64, 22.0±0.55 and 21.7±1.48 kg, respectively. The sheep were offered *ad lib* access to silage between 09:00am and 10:00am and a 250 g of concentrate mixture at 04:00pm daily. The concentrate mixture, consisting of groundnut cake, barley grain, wheat bran, common salt and mineral mixture in proportions of 40:30:27:2:1, respectively, had a crude protein of 170, neutral detergent fiber of 431.9, acid detergent fiber of 150, cellulose of 104 and lignin of 38.5 g/kg of dry matter. Following four weeks of feeding, a digestibility and metabolism trial was carried out. During this period, the sheep were housed in cages equipped for separate collection of faeces and urine. The animals had a two-day adaptation period, followed by five days of sample collection. Daily, individual animal faeces and urine were collected and pooled in iron trays and plastic containers, respectively. Representative samples of faeces for dry matter and nitrogen (N) estimation were collected for analyses. For urine, an aliquot of 1/100 was kept for N estimation in glass bottles and acidified with 5 ml of concentrated $H_2SO_4$. Daily collection of offered feed (silage and concentrate mixture) and orts were also collected daily and representative samples kept for dry matter estimation. The collected dry faecal samples, feed and orts were ground through a 1 mm sieve using a Willey mill and stored in plastic containers for chemical and biochemical evaluation.

At the end of the feeding trial, rumen liquor was collected from the animals using suction tubes before feeding (0 h). Approximately, 100 ml of a representative sample was drawn from

each sheep and strained through a double layer of muslin cloth. The rumen pH was measured immediately after collection with a digital pH meter (Systronic pH system 361). About 80 ml of rumen liquor was treated with a few drops of saturated mercuric chloride, frozen in labeled polypropylene bottles for metabolites estimation. The remaining portion of the rumen liquor was preserved in polypropylene bottles with 10% formalin for microbial counts.

**Rumen fermentations and microbial counts.** Protozoa and sporangia in formalin were quantified following the preserved rumen liquor method [24] while holotrichs and entodiniomorphs were identified using method of Ogimoto and Imai [25]. Metabolites in the rumen, such as total volatile fatty acids using steam distillation, total nitrogen using Kjeldahl digestions, and ammonia-nitrogen concentrations through microdiffusion technique were determined as described [26–28].

**Calculations for energy and digestible crude protein values and statistical analysis.** Total digestible nutrients and net energy values for lactation ($NE_L$), gain ($NE_G$) and maintenance ($NE_M$) of the silage samples were calculated using the following equations according to previous equations [29]:

Total digestible nutrients = 04.97- (1.302* acid detergent fiber)

Where: $NE_L$ = (total digestible nutrients *0.0245)-0.012

$NE_G$ = (total digestible nutrients *0.029)-1.01

$NE_M$ = (total digestible nutrients *0.029)-0.29)

Digestible Energy (KJ/g of dry matter) = total digestible nutrients *0.04409 according to previously described method [30].

Metabolizable Energy (KJ/g of dry matter) = 0.821 x digestible energy according to earlier method [31]. Digestible crude protein (digestible crude protein, g/kg of dry matter) = crude protein digested/feed intake × 1000

Where crude protein digested = crude protein intake (g)- crude protein voided in faeces (g).

In this study, we used completely randomized design for both green (fresh) and silage samples. In vitro data underwent one-way ANOVA, with genotypes/silage diets as fixed factors and other parameters as dependent variables. Our aim was to assess differences in chemical composition, carbohydrate and protein fractions, energy values, and silage fermentation quality (pH, lactic acid, and dry matter). For the *in vivo* study randomized block design was adopted, sheep data were subjected to a one-way analysis of variance where experimental diets were considered as the fixed factor, while estimated parameters (nutrients intake, digestibility, nitrogen balance, rumen parameters) served as a dependent variable. Significance among variable means was assessed at the *P < 0.05* level using the Duncan multiple-range test [32].

## Results

### Chemical composition of fresh material

There were significant variations (*P < 0.05*) in dry matter, organic matter and ether extract contents among the fresh *C. ciliaris* genotypes used for the silage study (Table 1). The crude protein also exhibited significant differences (*P < 0.05*), with IG96-358 recording the highest value at 91.6 g/kg of dry matter. The *C. ciliaris* genotypes displayed variations (*P < 0.05*) in neutral detergent fiber, acid detergent fiber, cellulose and lignin contents.

### Carbohydrate, protein fractions and energy value for fresh samples

Genotype IG96-96 had the lowest (*P < 0.05*) total carbohydrates and structural carbohydrates contents (771.7 and 710.2 g/kg of dry matter, respectively) and higher (*P < 0.05*) non-structural carbohydrates contents (61.5 g/kg of dry matter) compared with the other genotypes

**Table 1. Chemical composition and sugar contents (mg/g of dry matter) of fresh *Cenchrus ciliaris* genotypes.**

| Nutrients | IG96-401 | IG96-50 | IG96-96 | IG96-403 | IG96-358 | Standard error of means | *P* value |
|---|---|---|---|---|---|---|---|
| Dry matter | 389[c] | 329[a] | 350[b] | 329[a] | 349[b] | 11.27 | <0.001 |
| Organic Matter | 874[b] | 856[a] | 849[a] | 882[b] | 901[c] | 3.61 | <0.001 |
| Ether extract | 18.7[d] | 18.4[d] | 14.9[a] | 16.6[c] | 15.6[b] | 0.40 | <0.001 |
| crude protein | 56.2[a] | 56.8[a] | 68.5[b] | 58.1[a] | 91.6[c] | 3.15 | <0.001 |
| Neutral detergent fiber | 766[b] | 792[c] | 733[a] | 767[b] | 772[b] | 4.92 | <0.001 |
| Acid detergent fiber | 467[ab] | 487[b] | 516[c] | 455[a] | 452[a] | 6.24 | <0.001 |
| Cellulose | 367[b] | 362[b] | 385[c] | 347[a] | 357[ab] | 3.32 | <0.001 |
| Hemi cellulose | 299[b] | 305[b] | 215.8[a] | 312[b] | 319[b] | 9.50 | <0.001 |
| Lignin (sa) | 57.9[a] | 71.0[b] | 83.1[c] | 63.1[ab] | 69.3[b] | 2.24 | <0.001 |
| Sugar | 79.7[b] | 84.3[c] | 80.4[bc] | 73.7[a] | 78.1[b] | 3.78 | 0.031 |

[a,b,c,d]Means with different superscripts in a row differ significantly (*P < 0.05*).

(Table 2). Protein fractions $P_A$, $P_{B1}$, $P_{B2}$, $P_{B3}$ and $P_C$ differed (*P < 0.05*) among the genotypes and the $P_A$ fraction range (294.6–423.6 g/kg crude protein) was higher than that of $P_{B1}$ (131.0–258.6g/kg crude protein), $P_{B2}$, (91.2–172.5g/kg crude protein), $P_{B3}$ (87.4–156.1g/kg crude protein) and $P_C$ (175.2–226.0 g/kg crude protein) fractions. Energy values as total digestible nutrients, digestible energy and metabolizable energy were lower for IG96-96 (377.2 g/kg of dry matter, 6.95 and 5.71MJ/kg of dry matter, respectively) than the other genotypes. Similarly, net energy values for animal function viz. $NE_L$, $NE_M$ and $NE_G$ were lowest for IG96-96.

**Table 2. Carbohydrates, protein fractions and energy value of fresh five *Cenchrus ciliaris* genotypes.**

| Parameters | IG96-401 | IG96-50 | IG96-96 | IG96-403 | IG96-358 | Standard error of means | *P* value |
|---|---|---|---|---|---|---|---|
| *Carbohydrates* (g/kg of dry matter) | | | | | | | |
| tCHO | 799[c] | 808[c] | 772[a] | 802[c] | 787[b] | 3.61 | <0.001 |
| Non-structural CHO | 50.0[ab] | 35.0[a] | 61.5[c] | 54.5[c] | 45.0[ab] | 3.06 | 0.037 |
| Structural CHO | 749[b] | 772[c] | 710[a] | 748[b] | 742[b] | 5.69 | <0.001 |
| *Protein fractions* (g/kg crude protein) | | | | | | | |
| $P_A$ | 366[ab] | 380[ab] | 294 [b] | 423[b] | 403 [c] | 15.8 | 0.064 |
| $P_{B1}$ | 172[a] | 131[a] | 258[b] | 161[a] | 136 [a] | 15.6 | 0.034 |
| $P_{B2}$ | 172[c] | 168[bc] | 91.2[a] | 103[a] | 121[ab] | 10.5 | 0.009 |
| $P_{B3}$ | 87.4[a] | 97.9[ab] | 130[bc] | 137[bc] | 156[c] | 8.25 | 0.016 |
| $P_C$ | 201[ab] | 223[b] | 226[b] | 175[a] | 182[a] | 6.81 | 0.022 |
| *Energy value* (MJ/kg of dry matter) | | | | | | | |
| Digestible energy | 8.14[bc] | 7.65[b] | 6.95[a] | 8.43[c] | 8.48[c] | 0.164 | <0.001 |
| Metabolizable energy | 6.68[bc] | 6.28[b] | 5.71[a] | 6.92[c] | 6.96[c] | 0.135 | <0.001 |
| $NE_L$ | 4.02[bc] | 3.75[b] | 3.36[a] | 4.18[c] | 4.21[c] | 0.092 | <0.001 |
| $NE_G$ | 1.13[bc] | 0.81[b] | 0.35[a] | 1.32[c] | 1.36[c] | 0.108 | <0.001 |
| $NE_M$ | 4.14[bc] | 3.82[b] | 3.36[a] | 4.33[c] | 4.37[c] | 0.108 | <0.001 |
| Total digestible nutrients (g/kg of Dry matter) | 442[bc] | 415[b] | 377[a] | 457[c] | 460[c] | 8.92 | <0.001 |

[a,b,c]Means with different superscripts in a row differ significantly (P < 0.05), tCHO: total carbohydrate, CHO: carbohydrate, $P_A$: Non-protein nitrogen; $P_{B1}$: Buffer soluble protein; $P_{B2}$: Neutral detergent soluble protein; $P_{B3}$: Acid detergent soluble protein; $P_C$: Indigestible protein; $NE_L$: Net energy for lactation; $NE_M$: Net energy for maintenance; $NE_G$: Net energy for growth.

**Table 3. Fermentation quality of silages made using five *Cenchrus ciliaris* genotype.**

| Parameters (g/kg of dry matter) | IG 96–401 | IG 96–50 | IG 96–96 | IG 96–403 | IG 96–358 | Standard error of means | P value |
|---|---|---|---|---|---|---|---|
| Dry matter | 339[c] | 274[b] | 258[a] | 254[a] | 270[b] | 7.31 | <0.001 |
| pH | 4.06[a] | 4.61[b] | 4.97[d] | 4.91[c] | 5.17[e] | 0.089 | <0.001 |
| Lactic acid | 28.1[d] | 17.1[c] | 13.7[b] | 11.3[a] | 10.8[a] | 1.47 | <0.001 |
| $NH_3$-N | 0.39[a] | 0.50[a] | 0.71[b] | 0.73[b] | 0.91[b] | 0.049 | 0.001 |

[a,b,c,d,e]Means with different superscripts in a row differ significantly (P < 0.05), $NH_3$-N: Ammonia nitrogen.

## Fermentation quality and chemical composition of silage

*C. ciliaris* silages differed (*P < 0.05*) in dry matter (254-339g/kg of dry matter), pH (4.06–5.17) and lactic acid contents (10.8–28.0 g/kg of dry matter) as indicated in Table 3. Ammonia-N concentration ranged between 0.39 to .91 g/kg and significant (*P < 0.05*) differences were noted among the different *C. ciliaris* genotypes.

The Organic Matter, ether extract and hemicelluloses contents of maize silage were higher (*P < 0.05*) than IG96-401 and IG96-96 silages (Table 4). Fermentation quality of silages (IG96-401, IG96-96 and maize) fed to sheep was significantly higher (*P < 0.05*) for dry matter, pH and lactic acid contents.

## Carbohydrate, protein fractions and energy value of silages

The total carbohydrates and non-structural carbohydrates contents of maize silage were significantly higher (*P < 0.05*), while structural carbohydrates content was comparable (*P < 0.05*) with *C. ciliaris* silages (Table 5). Protein fractions $P_A$, $P_{B1}$ and $P_C$ differed (*P < 0.05*) amongst the silages. The calculated values of digestible dry matter (DDM) and relative feed value (RFV) were lower (*P < 0.05*) for IG96-96 (465.5g/kg of dry matter and 55.3%). The total digestible nutrients, digestible energy, metabolizable energy, $NE_M$, $NE_G$ and $NE_L$ contents were lower (*P < 0.05*) for IG96-96.

## *In vivo* digestibility of sheep

Intakes dry matter, crude protein, digestible crude protein and metabolizable energy, g/$kg^{w0.75}$) by the sheep were significantly different (*P < 0.05*) in the silage diets (Table 6).

**Table 4. Chemical composition, pH and lactic acid of silages fed to sheep.**

| Parameters (g/kg of dry matter) | *C. ciliaris* Genotypes silages | | Maize silage | Standard error of means | *P* value |
|---|---|---|---|---|---|
| | IG96-401 | IG96-96 | | | |
| Dry matter | 369.3 [b] | 295.5 [a] | 357.9 [b] | 14.51 | <0.047 |
| Organic Matter | 865[a] | 881[b] | 919.5[c] | 8.27 | <0.001 |
| Crude protein | 58.6 | 70.7 | 59.4 | 2.61 | 0.085 |
| Ether extract | 17.6[b] | 15.1[a] | 20.4[c] | 0.78 | <0.001 |
| Neutral detergent fiber | 764.8 | 782.6 | 780.1 | 4.74 | 0.284 |
| Acid detergent fiber | 505.4[a] | 543.6[b] | 507.8[a] | 7.74 | 0.060 |
| Cellulose | 366.7 | 397.9 | 402.0 | 7.45 | 0.086 |
| Lignin | 76.5 | 84.7 | 76.7 | 1.79 | 0.080 |
| Hemi cellulose | 257.9[b] | 239.0[a] | 272.3[c] | 4.98 | <0.001 |
| pH | 4.41 [a] | 4.86 [b] | 4.79 [b] | 0.05 | <0.001 |
| Lactic acid | 31.1 [b] | 20.9 [a] | 23.7 [a] | 1.33 | 0.021 |

[a,b,c]Means with different superscripts in a row differ significantly (P < 0.05).

**Table 5. Carbohydrates, protein fractions and energy value of silages fed to sheep.**

| Parameters | *C. ciliaris* genotypes silages | | Maize silage | Standard error of means | *P* value |
|---|---|---|---|---|---|
| (g/kg of dry matter) | IG96-401 | IG96-96 | | | |
| *Carbohydrates* | | | | | |
| tCHO | 788.8[a] | 794.8[a] | 839.8[b] | 4.88 | <0.001 |
| Non-structural CHO | 45.8[a] | 38.6[a] | 87.0[b] | 7.52 | 0.001 |
| Structural CHO | 743.0 | 756.4 | 755.4 | 4.47 | 0.452 |
| *Protein fractions* | | | | | |
| $P_A$ | 247.0[a] | 328.5[b] | 384.0[c] | 20.4 | <0.001 |
| $P_{B1}$ | 371.2[b] | 226.1[a] | 210.6[a] | 26.4 | <0.001 |
| $P_{B2}$ | 44.3 | 70.0 | 41.2 | 6.93 | 0.183 |
| $P_{B3}$ | 80.5 | 63.9 | 91.7 | 6.63 | 0.249 |
| $P_C$ | 257.0[a] | 311.5[b] | 272.5[c] | 8.18 | <0.001 |
| *Energy values* | | | | | |
| Digestible energy | 7.18[b] | 6.30[a] | 7.16[b] | 0.186 | 0.062 |
| Metabolizable energy | 5.90[b] | 5.17[a] | 5.88[b] | 0.153 | 0.060 |
| $NE_L$ | 3.49[b] | 3.00[a] | 3.48[b] | 0.103 | 0.059 |
| $NE_G$ | 0,50[b] | -0.10[a] | 0.49[b] | 0.022 | 0.061 |
| $NE_M$ | 3.51[b] | 2.93[a] | 3.50[b] | 0.122 | 0.059 |
| Total digestible nutrients | 389.7[b] | 342.0[a] | 388.5[b] | 10.1 | 0.060 |

[a,b,c]Means with different superscripts in a row differ significantly (*P < 0.05*), tCHO: total carbohydrate, CHO: carbohydrate, $P_A$: Non-protein nitrogen; $P_{B1}$: Buffer soluble protein; $P_{B2}$: Neutral detergent soluble protein; $P_{B3}$: Acid detergent soluble protein; $P_C$: Indigestible protein; $NE_L$: Net energy for lactation; $NE_M$: Net energy for maintenance; $NE_G$: Net energy for growth.

Nutrients digestibility except for acid detergent fiber and cellulose was similar (*P > 0.05*) in sheep fed the silage diets. Nitrogen intake, urinary-N excretion and N retention in sheep was higher (*P < 0.05*) in IG96-96 silage diet than in maize and IG96-401 silage diets. The digestible energy and metabolizable energy contents of silage diets were similar, while digestible crude protein (%) content of maize silage diet was lower (*P < 0.05*) than in *C. ciliaris* silage diets.

## Rumen fermentation and microbial counts

The pH, rumen metabolites (total volatile fatty acids, total-N and $NH_3$-N) and microbial counts of (total protozoa, entodinio morphs, holotrich and sporangia; Table 7) were comparable (P>0.05) in the rumen liquor of sheep fed the silage-based diets.

# Discussion

## Chemical composition of fresh material

Proteins are building blocks of life. Crude protein is therefore needed in animal nutrition for the growth and regeneration of tissues. The crude protein content which is an indicator of nutritional quality [33], was lower than generally recommended for the animal in *C. ciliaris* silages. Optimal levels (16–18%) of crude protein are essential in both forage and preserved feed such as silage [34]. In instance where the crude protein percentage is low, the rumen microflora responsible for digestion struggle to maintain their population at adequate levels to effectively process the feed [35]. Except for genotype IG97-358, which recorded crude protein of 91.6 g/kg of dry matter, the values observed in the present study were less than the 70.0 g/kg of dry matter required for sustaining rumen microbial growth [36]. Similar to findings in the current study, in a previous study [37], they reported values of 881–904, 80–96, 687–738, 485–

**Table 6. Nutrient intake, digestibility, nitrogen balance and nutritive value of the silage diets in sheep.**

| Parameters | *C. ciliaris* genotypes silages | | Maize silage | Standard error of means | *P* value |
|---|---|---|---|---|---|
| | IG96-401 | IG96-96 | | | |
| Animal weight (kg) | 22.53 | 22.48 | 22.73 | 0.706 | 0.990 |
| *Nutrients intakes* | | | | | |
| Dry matter (g/day) | 626.55[a] | 723.97[ab] | 764.34[b] | 24.83 | 0.047 |
| Dry matter (% body weight) | 2.79[a] | 3.22[b] | 3.37[b] | 0.082 | <0.001 |
| Dry matter (g/kg$^{w0.75}$) | 60.67[a] | 70.15[b] | 73.52[b] | 1.735 | <0.001 |
| Crude protein (g/day) | 66.20[a] | 74.62[b] | 67.65[a] | 1.537 | 0.355 |
| Crude protein (g/kg$^{w0.75}$) | 6.44[a] | 7.23[b] | 6.53[a] | 0.128 | 0.005 |
| Digestible crude protein (g/day) | 33.43[ab] | 37.85[b] | 30.27[a] | 1.478 | 0.071 |
| Digestible crude protein (g/kg$^{w0.75}$) | 1.99[ab] | 2.54[b] | 1.79[a] | 0.151 | 0.096 |
| Metabolizable energy (MJ/day) | 5.35[a] | 5.87[ab] | 6.66[b] | 0.228 | 0.043 |
| Metabolizable energy (MJ/kg$^{w0.75}$) | 0.521[a] | 0.570[ab] | 0.640[a] | 0.018 | 0.011 |
| *Nutrient digestibility (g/kg of dry matter)* | | | | | |
| Dry matter | 524 | 492 | 533 | 12.2 | 0.390 |
| Organic Matter | 552 | 532 | 557 | 11.18 | 0.661 |
| Crude protein | 507 | 507 | 447 | 16.23 | 0.241 |
| Neutral detergent fiber | 491 | 474 | 523 | 13.95 | 0.378 |
| Acid detergent fiber | 428[ab] | 406[a] | 462[b] | 19.26 | 0.046 |
| Cellulose | 501[a] | 533[b] | 512[ab] | 25.89 | 0.037 |
| *Nitrogen Balance (g/day)* | | | | | |
| Nitrogen intake | 10.59[a] | 11.94[b] | 10.82[a] | 0.246 | 0.037 |
| Faecal nitrogen | 5.09 | 5.88 | 5.98 | 0.240 | 0.275 |
| Urinary nitrogen | 1.47[a] | 2.20[b] | 1.37[a] | 0.156 | 0.012 |
| N absorbed | 5.50[ab] | 6.06[b] | 4.84[a] | 0.239 | 0.109 |
| N retention | 4.04 | 3.75 | 3.47 | 0.204 | 0.690 |
| *Nutritive value* | | | | | |
| Digestible energy (MJ/kg of dry matter) | 10.44 | 9.90 | 10.61 | 0.216 | 0.391 |
| Metabolizable energy (MJ/kg of dry matter) | 8.57 | 8.11 | 8.69 | 0.175 | 0.393 |
| Digestible crude protein (%) | 5.40[b] | 5.22[b] | 3.98[a] | 0.251 | 0.021 |

[a,b]Means with different superscripts in a row differ significantly (*P < 0.05*).

**Table 7. Rumen metabolites and microbial counts of rumen liquor of sheep fed different silages.**

| Parameters | *C. ciliaris* genotypes silages | | Maize silage | Standard error of means | *P* value |
|---|---|---|---|---|---|
| | IG96-401 | IG96-96 | | | |
| pH | 6.66 | 6.86 | 6.69 | 0.049 | 0.202 |
| Total volatile fatty acids (meq/L) | 85.62 | 80.50 | 87.25 | 6.514 | 0.923 |
| $NH_3$-N (mg/100ml) | 23.33 | 25.23 | 22.4 | 1.557 | 0.725 |
| Total-N (mg/100ml) | 56.12 | 63.72 | 57.89 | 4.721 | 0.541 |
| *Microbial counts (×10$^5$/ml)* | | | | | |
| Total protozoa | 2.56 | 3.12 | 3.31 | 0.203 | 0.321 |
| Holotrichs | 0.32 | 0.39 | 0.41 | 0.084 | 0.231 |
| Entodiniomorphs | 2.24 | 2.73 | 2.90 | 0.171 | 0.872 |
| Fungal sporangia | 0.25 | 0.21 | 0.29 | 0.065 | 0.243 |

$NH_3$-N: Ammonia Nitrogen; N: Nitrogen

519, 367–432 and 34–60 g/kg for Organic Matter, crude protein, neutral detergent fiber, acid detergent fiber, cellulose and lignin, respectively, in five other *C. ciliaris* genotypes. The mean values for Organic Matter, crude protein, neutral detergent fiber, cellulose and lignin for 78 *C. ciliaris* genotypes evaluated in Mexico [38] were 861, 82, 734, 413 and 31 g/kg of dry matter, respectively. Additionally, the values for Organic Matter, crude protein, ether extract, neutral detergent fiber, acid detergent fiber, cellulose and lignin were 873, 41, 19.5, 682.3, 418.7, 360 and 55 g/kg of dry matter, respectively, for *C. ciliaris* [39], which fell within the range observed in our present study.

## Carbohydrate, protein fractions and energy value for fresh samples

Dry matter content of plants is made of up to 700 to 800 g/kg of dry matter of carbohydrate an important source of energy for rumen microorganisms [36]. Higher non-structural carbohydrates (61.5 g/kg of dry matter) for IG96-96 genotype may be due to its lower neutral detergent fiber and hemicellulose contents (734 and 215g/kg of dry matter) as neutral detergent fiber contents of forage may vary with carbohydrate levels [40]. The non-structural carbohydrates content of 42.0 g/kg of dry matter recorded for *C. ciliaris* [41] was similar to values obtained for this study. Additionally, the mean total carbohydrates values of 730–836 g/kg of dry matter for seven tropical grasses cut at 56 days after sowing [42] were equivalent to values of 772–808 g/kg of dry matter gained in the present study. The total carbohydrates and Non fiber carbohydrate (NFC) contents were recorded between 728–827 and 20 to 90 g/kg of dry matter, respectively for *Cynodon dactylon*, *Brachiaria brizantha* and *Megathyrsus maximus* [43].

The *C. ciliaris* genotypes evaluated in the present study had higher $P_C$ contents (175–226 g/kg of crude protein) than the normal range of about 50–150 g/kg of crude protein. Higher amounts of the $P_C$ fraction is undesirable in any ingredient or diet as this fraction is bound to lignin and unavailable to rumen microflora [36]. Higher $P_C$ values for IG96-96 may be partly attributed to its higher acid detergent fiber and lignin contents of 358 and 83.1 g/kg of dry matter, respectively. The values of 346, 15.2, 137, 19.5 and 480 g/kg of crude protein reported [41] for $P_A$, $P_{B1}$, $P_{B2}$, $P_{B3}$ and $P_C$ contents, respectively, for *C. ciliaris* were not equivalent to corresponding protein fraction values recorded in the present study. An another study [44] reported significant (*P < 0.05*) differences in protein fractions in grass species and their harvest age. Protein fraction $P_A$ was lower in *Andropogon gayanus Kunth* (120–130 g/kg of crude protein) than in *C. ciliaris* and *Panicum maximum* cv. Massai with160-170 g/kg of crude protein when cut at day 63. The above grasses $P_{B2}$, $P_{B3}$ and $P_C$ protein fractions ranged from 280–340, 270–310 and 213-2736g/kg of crude protein, respectively, when cut at day 63. Another study [45] reported protein fractions $P_A$, $P_{B1}$, $P_{B2}$, $P_{B3}$ and $P_C$ of *Jiggs Bermuda* grass in (fall, winter, spring and summer) of between 408–550, 138–139, 101–157, 125–148 and 80.2-115g/kg of crude protein, respectively which is consistent with values in the present study.

The *C. ciliaris* genotypes had an inadequate energy range of 5.71–6.96 KJ/g of dry matter to meet the maintenance requirements of 8.32 KJ/g of dry matter metabolizable energy (ME) recommended for ruminants, as reported [2]. This suggests that the *C. ciliaris* genotypes alone may not be sufficient as a sole feed for ruminants, necessitating supplementation to address energy deficits. The total digestible nutrients (377.2–460.3), digestible energy (6.95–8.48) and ME (5.71–6.96 KJ/g of dry matter) calculated for *C. ciliaris* genotypes in the present study fall within the range of 342–609 g/kg of dry matter, 5.92–11.26 and 4.85–9.23 MJ/kg of dry matter, respectively, for ten tropical grasses [46]. The higher metabolizable energy contents of *C. ciliaris* (7.65–9.02 MJ/kg of dry matter) at 60 to 120 days of crop growth was reported [47]. The metabolizable energy contents of between 7.7–13.6 MJ/kg of dry matter [48] were also higher than the metabolizable energy values recorded for *C. ciliaris* in this study. The lowest

total digestible nutrients and digestible energy values of IG96-96 could be attributed to its higher acid detergent fiber and lignin contents (517 and 83.1 g/kg of dry matter) as higher accumulation of acid detergent fiber and lignin reduces nutrients utilization in forages [49]. The $NE_L$ values (3.36–4.21MJ/kg of dry matter) fall within the values for Napier (*Pennisetum purpureum*) and Pangola grass (*Digitaria eriantha*) (4.38 and 4.85 MJ/kg of dry matter) [50]. The $NE_M$ values of *C. ciliaris* genotypes recorded in the present study (3.36–4.37 MJ/kg of dry matter) are inadequate for mature beef cattle, which require between 4.92 and 5.30 MJ/kg of dry matter [51].

## Silage fermentation quality

The dry matter contents of *C. ciliaris* silages of 255–339 g/kg of dry matter fell within the typical range for grass silages. The pH of silage in a crop is governed by various plant characteristics, such as soluble sugar content, buffering capacity, endophytic lactic acid producing bacteria, and dry matter content amongst other parameters [52]. Fermentation during silage production helps to prevent growth of spoilage bacteria and to preserve forage in the low oxygen environment. Silage made from the *C. ciliaris* genotypes had pH values between 4.06 (IG96-401) to 5.17 (IG97-358) and were less acidic than the reported pH 3.8 to 4.2 ranges [53] for corn/sorghum/oat silages. A pH of 3.8 to 4.2 occurs in silage that is well prepared and will keep well. Silage with elevated pH (exceeding 4.2) tends to have poor preservation and is often associated with the higher ammonia levels [54]. On the other hand, pH values of 3.6 and below are deemed too acidic and may lead to stomach upsets in animals when included in their diet. A well-made grass silage is recommended to have a pH lower than 4.47, along with lactic acid levels ranging between 40 to 70 g/kg of dry matter [55]. The pH values recorded for *C. ciliaris* silages were acceptable and consistent with some previously reported values for grasses [56, 57]. Another study reported a pH range of 4.6–5.4 for *C. ciliaris* grass silage after 30 days of fermentation [58]. After 60 days of ensiling at temperature of 28˚C, *Paspalum plicatulum* grass exhibited a pH of 5.2 and a lactic acid content of 17 g/kg of dry matter, while at an ensiling at temperature of 40˚C it had a pH of 5.1 and a lactic acid content of 20 g/kg of dry matter [59]. It was also found that silage of the tropical grass *Pennisetum purpureum* exhibited a higher pH of 5.45 and a lower lactic acid concentration of 9.0 g/kg of dry matter [60] when compared to the pH and lactic acid levels observed in the *C. ciliaris* genotype silages. On the contrary, the lower pH of 4.04–4.47 and higher lactic acid range of 39.1–76.5 g/kg of dry matter for six warm season grasses was recorded [61]. The $NH_3$-N values in the fresh materials of native grasses ranged from 0.51 to 0.72 g/kg [62], and this was consistent with the obtained values except for genotype IG97-358. An another study [63] noted an $NH_3$-N level of 0.47 g/kg in fresh Inner Mongolian grass, which was lower than the corresponding values of 0.71, 0.73 and 0.91 g/kg in fresh materials for genotype IG96-96, IG96-403 and IG97-358, respectively, recorded in this study.

## Chemical composition of silages fed to sheep

Following the fermentation quality assessment, three out of the five genotypes (IG 96–50, IG 96–401, and IG 96–96) were deemed suitable for ensiling. However, due to the unavailability of material for IG 96–50, we proceeded with only these two genotypes (IG 96–401 and IG 96–96) for further in vivo feeding trials. The crude protein content in the silages provided to the sheep was generally below the minimum requirement of 70 g/kg of dry matter for the growth of rumen microflora, except for genotype IG96-96, which had a content of 70.7 g/kg of dry matter [36]. The acid detergent fiber and lignin contents of IG96-401 and IG96-96 silages were higher, while hemicellulose was lower compared to green/fresh forage. This discrepancy may

be attributed, in part, to the Maillard reaction, an organic chemical reaction in which reducing sugars react with amino acids during ensiling and form a complex mixture of compounds. Others observed significantly lower ($P < 0.05$) hemicellulose and cellulose contents in silage compared to hay and fresh grass [55]. The ether extract contents in *C. ciliaris* silages ranged from 15.1 to 17.6 g/kg of dry matter. The contents were relatively lower than those in green/fresh forage (18.4–18.7 g/kg of dry matter). The crude protein and fiber contents of *C. ciliaris* silage fall within the reported range of values for crude protein (34.0–109 g/kg of dry matter), neutral detergent fiber (570–693 g/kg of dry matter) and, acid detergent fiber (393–460 g/kg of dry matter) for nine warm season grasses harvested in summer [64]. Previous studies [9, 60, 65] documented reduced hemicellulose contents upon ensiling.

## Carbohydrate, protein fractions and energy value of silages fed to sheep

The total carbohydrates contents in *C. ciliaris* silages were relatively lower than values observed fresh forage. This reduction in total carbohydrates for *C. ciliaris* could be attributed to the energy required from carbohydrates for various metabolic activities of microflora to facilitate effective fermentation. Similar findings were reported, where total carbohydrates, $C_A$, $C_{B1}$ and $C_{B2}$ fractions decreased ($P < 0.05$) in ensiled native tropical and temperate grass grown in Inner Mongolia [57]. In the present study, total carbohydrates and non-structural carbohydrates contents were higher in maize silage (839.8 and 87.0 g/kg of dry matter) than for *C. ciliaris* genotypes IG96-401 (788.8 and 45.8, g/kg of dry matter) and G96-96 (794.8and 38.6 g/kg of dry matter), respectively. A study [4] reported that tropical grasses usually had lower non-structural carbohydrates concentration than did corn and sorghum. Corn silage total carbohydrates of 856.0 and 825.9 g/kg of dry matter reported earlier [66, 67] were similar to the total carbohydrates contents of maize silage in the present study, however the non-structural carbohydrates/NFC contents for corn silage (39.4 and 44.03 g/kg of dry matter) were lower than the current value (87.0 g/kg of dry matter).

A higher $P_C$ fraction in IG96-401 and IG96-96 silages compared to the fresh forage could be due to Millard products forming during ensiling, as reported previously [36]. Similarly, others [68] reported increased $P_{B3}$ and $P_C$ due to ensiling, but in the present study, the $P_{B3}$ fraction did not increase on ensiling. The $P_{B2}$ fraction of *C. ciliaris* silages decreased due to ensiling, similar to earlier observations [63] that reported a decrease ($P<05$) in $P_{B2}$ fraction in a native grass (*Stipa grandis*, *Leymus chinensis*, *Serratula centauroides* L. *Astragalus melilotoides* Pall, *Stipa krylovii*, *Cleistogenes squarrosa*, and *Artemisia frigida*) after ensiling.

The lower energy values observed in this study may be attributed to the higher acid detergent fiber contents found in genotypes IG96-401, IG96-96, and maize silages.

## *In vivo* digestibility by sheep

The dry matter intake (DMI) of forage is primarily determined by its fibre concentration, digestibility and degradation rate in the rumen [69]. Lower DMI for genotype IG96-401 silage diet in sheep could be attributed to its lower Organic Matter and higher ash contents. Higher digestible crude protein and metabolizable energy intake in sheep on genotype IG96-96 and maize silage diets could be ascribed to their high crude protein and metabolizable energy contents. Sheep metabolizable energy intake recorded in the present study was lower than metabolizable energy intake of 7.43–11.96 MJ/d in sheep fed ryegrass silage and ryegrass silage supplemented with barley grain [70]. The dry matter, Organic Matter, neutral detergent fiber and acid detergent fiber digestibility of sheep fed elephant grass silage [71] resembled like values in the present study with a lower crude protein digestibility. Differences in acid detergent

fiber and cellulose digestibility of silage diets in sheep may be attributed to differences in their acid detergent fiber and cellulose contents.

There is a direct relationship between dietary protein level and the amount of N excreted in faeces. Nitrogen retention in animals depends on its excretion through faeces and urine. All the sheep in the present study had positive N balance. This shows that *C. ciliaris* silages will be able to meet and exceed the maintenance requirements of the animals [40]. Lower digestible crude protein content of maize silage diet could be attributed to its lower crude protein digestibility (448 g/kg of dry matter) noted in the present study.

### Rumen metabolites and microbial counts

The comparatively elevated $NH_3$-N levels in the rumen liquor of sheep fed a diet based on genotype IG96-96 silage might be a result of higher intake of crude protein. Previous studies [72, 73] also reported a positive correlation between N intake and rumen $NH_3$-N concentration. Higher protozoa count in sheep fed on the maize silage could be attributed to higher non-structural carbohydrates of maize silage. Protozoa grow constantly and have larger population on a diet rich in readily available carbohydrate and are known to play a significant role in plant carbohydrate breakdown in the gastro-intestinal tract [74].

### Conclusions

*C. ciliaris* genotypes exhibited significant differences in protein, fiber contents, protein fractions and energy contents. The silages derived from the *C. ciliaris* genotypes were of good to medium quality. Notably, *C. ciliaris* genotypes such as IG96-401 and IG96-96 exhibited feeding values comparable to maize silage across various parameters, including intakes, nutrients utilization, nitrogen balance, nutritive value, and rumen fermentation. Looking ahead, exploring and optimizing the characteristics of *C. ciliaris* genotypes could offer promising alternatives to maize silage, helping the frame for future landscape of livestock nutrition and silage quality.

### Supporting information

**S1 File. Supplementary data: Replication dataset for final tables.**
(PDF)

### Acknowledgments

Authors appreciate the support from the Director of ICAR-IGFRI, Jhansi for facilitating this research work.

### Author Contributions

**Conceptualization:** Pushpendra Koli, Madan Mohan Das, Subhendu Bikash Maity, Krishna Kunwar Singh, Rohit Katiyar, Sanat Kumar Mahanta, Manoj Kumar Srivastava, YongLin Ren.

**Data curation:** Pushpendra Koli, Tejveer Singh.

**Formal analysis:** Tejveer Singh, Madan Mohan Das, Subhendu Bikash Maity, Krishna Kunwar Singh.

**Funding acquisition:** YongLin Ren.

**Investigation:** Asim Kumar Misra.

**Resources:** YongLin Ren.

**Supervision:** Sultan Singh, Asim Kumar Misra, YongLin Ren.

**Validation:** Pushpendra Koli, Krishna Kunwar Singh, Olatunde Akeem Oderinwale.

**Visualization:** Sanat Kumar Mahanta.

**Writing – original draft:** Pushpendra Koli, Krishna Kunwar Singh, Rohit Katiyar, Manoj Kumar Srivastava, Uchenna Young Anele, Olatunde Akeem Oderinwale, YongLin Ren.

**Writing – review & editing:** Pushpendra Koli, Manoj Kumar Srivastava, Uchenna Young Anele, YongLin Ren.

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
