## [Decision Letter · Decision Letter 0]

3 Apr 2024

PONE-D-24-07834Assessing genotypes of buffel grass (Cenchrus ciliaris) as an alternative to maize silage for sheep nutritionPLOS ONE

Dear Dr. Koli,

Thank you for submitting your manuscript to PLOS ONE. After careful consideration, we feel that it has merit but does not fully meet PLOS ONE’s publication criteria as it currently stands. Therefore, we invite you to submit a revised version of the manuscript that addresses the points raised during the review process.

We look forward to receiving your revised manuscript.

Kind regards,

Adham A. Al-Sagheer

Academic Editor

PLOS ONE

Journal Requirements:

Reviewers' comments:

Reviewer's Responses to Questions

**Comments to the Author**

1. Is the manuscript technically sound, and do the data support the conclusions?

Reviewer #1: Yes

Reviewer #2: Yes

2. Has the statistical analysis been performed appropriately and rigorously? 

Reviewer #1: Yes

Reviewer #2: No

3. Have the authors made all data underlying the findings in their manuscript fully available?

Reviewer #1: Yes

Reviewer #2: Yes

4. Is the manuscript presented in an intelligible fashion and written in standard English?

Reviewer #1: Yes

Reviewer #2: Yes

5. Review Comments to the Author

**Reviewer #1: **

After reviewing the paper “Assessing genotypes of buffel grass (Cenchrus ciliaris) as an alternative to maize silage for sheep nutrition," The research idea is excellent, particularly because it addresses the issue of feed shortages in animal production. Furthermore, it discusses an important point: which breeding varieties can be ensiled without the use of silage additives. The authors make several interesting points, making it difficult for the reader to follow their main argument. I recommend that the authors identify their main argument and structure their manuscript accordingly.

Materials & Methods

1. Material and method are unorganized, author should be organized as follows:

2.1. Experiment 1: Screening assays for different genotypes Cenchrus ciliaris

2.1.1. Fresh raw materials

The Experimental location and establishment section is listed

2.1.2. Silage perpration

The author should list the method of preparing silage in the laboratory for five types of feed Following that, it should mention how silage is prepared for sheep in vivo digestibility.

2.1.3. Fermentation profiles and chemical analysis

• The author should describe the steps taken to prepare the silage extract, which is estimated in terms of silage pH and lactic acid levels. In terms of lactic acid estimation, we must shorten this wholesale preparation method with lactic acid, which was determined using a spectrophotometric method as described in (reference).

• The chemical composition of fresh and silage, with a brief mention of sugar contents, should be written as follows: Sugar content was determined using... according to (reference). Additionally, include carbohydrate and protein fractions in this section.

2.2. Experiment 2: Invivo digestibility by sheep

- The corn silage used has no memory of its origins or whether it worked with the experiment treatment.

- Rumen metabolites should be referred to as rumen fermentations, and line 224 should be moved to line 227, following line 145.

- The method for estimating volatile fatty acids, ammonia, and nitrogen is not mentioned or referenced. For example, when estimating VFA concentration, did you evulate GC?

2. Some errors must be reconsidered; for example, the author estimates dry matter at 100 °C for 72 hours. It is well known that the relative humidity of air-dried materials is estimated at temperatures of 105°C for six hours, 125°C for four hours, or 135°C for three hours, whereas the total humidity of juicy and green materials is estimated at 40-60°C for 24-72 hours. Furthermore, the estimate of total carbohydrates is supposed to be called nitrogen-free extract; it is known that it is estimated at 100- (moisture + protein + fat + fiber + ash), and non-carbohydrate fiber at 100- (moisture + protein + fat + NDF + ash). This is something that the author must correct.

3. Calculations for energy and digestible crude protein values must be combined with statistical analysis.

Results

1. Does Table 1 show the chemical composition of a fresh or silage sample?

It should be clarified in the title of the table and the text, even if it is a fresh sample, where the chemical analysis of the silage sample is, and vice versa?

The same trend with carbohydrates, protein fractions, and energy values in Table 2. Chemical analysis is required for five types of samples, whether silage or fresh. Why does the author review the silage analysis of selected genotype results again in the in vivo assessment? It is a salvation that was discussed during the initial evaluation. I recommend removing it from the in vivo evaluation. Thus, I recommend that the author organize the results as follows:

• Chemical composition of fresh material

• Fermentation quality and chemical composition of silage

• In vivo digestibility by sheep

2. To combine silage fermentation characteristics and chemical analysis, it is advisable to present them in a single table. When writing about materials and methods, arrange them in the order of the table. It is preferable to start with the fermentation characteristics of silage and then include the chemical analysis in a table and write them in a manuscript. This approach will facilitate the interpretation of the results.

3. The results of the evaluation silage of five genotypes show that silage with IG 96-401 had the highest lactic acid, lowest ammonia concentration, and lowest pH. It was preferable to test it in vivo exclusively with maize silage. On what basis did the author select 96-96? Although the silage evaluation indicates that it is the best IG 96-50 after IG 96-401.

4. Why does the author include corn silage in the initial assessment of five types, especially since his main goal is to use it as an alternative feed to corn silage? His genotype selection was supposed to be based on the closest assessment of corn silage as the standard.

Discussion

1. In the section of the discussion, the results must be discussed as follows:

• Chemical composition of fresh material

The author should review studies that agree or disagree with the study of chemical composition of fresh sample.

• Fermentation quality and chemical composition of silage

- As you discuss the fermentation parameters for silage, you must clarify whether each parameter in the study conforms to the specifications of good silage or not. For example, if the values of the PH of silage, lactic acid, and ammonia for the standared (Maize silage) and genotypes are recommended for the good silage and indicative of its increase or decrease than the values recommended, you must discuss, with references. Furter, auther sould discuss change chemical composition of silage (labotary ensiling).

- The author should discuss the reasons for his choice of the two types of genotypes for animal evaluation.

• In vivo digestibility by sheep

- It should be noted that if there are results that do not have an impact, such as in nutrient digestion, for example, it is essential to refer to references that either agree or disagree with the study and explain the reasons behind this discrepancy.

- The author cites plants in the discussion that have nothing to do with the plant under study (underlines 459, 460, 464, and 465); however, he must also cite references to the plant under study.

**Reviewer #2: **

Line 96-967. Where all the samples harvested in the same day or in different days?

Line 109. Please explain why the authors used these two genotypes (IG96-96 and IG96-401) for the sheep study.

Line 229. Please be more specific what kind of experimental design the authors used for both green (fresh) and silage samples, and for the in vivo study.

Line 265 Table 2. PB1 of genotype IG96-50= 1310? I think is a typing error.

Line 310 Table 6. Please report daily weight gain, if possible, this data will help the readers to visualize the effect of feeding these type of genotypes

6. PLOS authors have the option to publish the peer review history of their article (what does this mean?). If published, this will include your full peer review and any attached files.

Reviewer #1: **Yes: **

Reviewer #2: No

---

## [Author Response · Author response to Decision Letter 0]

25 Apr 2024

Dear Editor, 

We would like to thank you for all reviewers for their highly insightful comments, which significantly helped me in the improvisation of the quality of our article. In the following pages, I have written the responses point by point for each comment of reviewers. Hopefully, I have incorporated reviewers all comments and advice.

Thanks, and kind regards,

Pushpendra

Response to Reviewers

Reviewer# 1

After reviewing the paper “Assessing genotypes of buffel grass (Cenchrus ciliaris) as an alternative to maize silage for sheep nutrition," The research idea is excellent, particularly because it addresses the issue of feed shortages in animal production. Furthermore, it discusses an important point: which breeding varieties can be ensiled without the use of silage additives. The authors make several interesting points, making it difficult for the reader to follow their main argument. I recommend that the authors identify their main argument and structure their manuscript accordingly.

Response: Thanks for your in-depth review and critical suggestions to improve our manuscript.

Materials & Methods

1. Material and method are unorganized, author should be organized as follows:

2.1. Experiment 1: Screening assays for different genotypes Cenchrus ciliaris

2.1.1. Fresh raw materials

The Experimental location and establishment section is listed

2.1.2. Silage perpration

The author should list the method of preparing silage in the laboratory for five types of feed Following that, it should mention how silage is prepared for sheep in vivo digestibility.

Response: The above suggestions are incorporated in the manuscript. Outline and subheadings are formed as suggested. 

2.1.3. Fermentation profiles and chemical analysis

• The author should describe the steps taken to prepare the silage extract, which is estimated in terms of silage pH and lactic acid levels. In terms of lactic acid estimation, we must shorten this wholesale preparation method with lactic acid, which was determined using a spectrophotometric method as described in (reference).

Response: The above suggestions are incorporated in the manuscript and reference also incorporated in the text.

• The chemical composition of fresh and silage, with a brief mention of sugar contents, should be written as follows: Sugar content was determined using... according to (reference). Additionally, include carbohydrate and protein fractions in this section.

Response: The above suggestions are incorporated with modification of text. The suitable reference also incorporated in the text.

2.2. Experiment 2: In vivo digestibility by sheep

- The corn silage used has no memory of its origins or whether it worked with the experiment treatment.

Response: Yes, maize is considered as one of the treatments along with Cenchrus genotypes during the silage feeding experiments. The values are given in table 4 and accordingly reported and discussed in respective sections. 

- Rumen metabolites should be referred to as rumen fermentations, and line 224 should be moved to line 227, following line 145. 

Response: The word rumen metabolites replaced by rumen fermentation. Sentence rearranged as per the suggestions.

- The method for estimating volatile fatty acids, ammonia, and nitrogen is not mentioned or referenced. For example, when estimating VFA concentration, did you evulate GC?

Response: Thanks for notifying this, we did not use GC here because we have not assessed individual component of volatile fatty acids, we focused on total VFAs. They are mentioned with the reference 26, 27 and 28 are described volatile fatty acids, ammonia, and total nitrogen respectively. The sentence also rephrases by incorporation important information.

2. Some errors must be reconsidered; for example, the author estimates dry matter at 100 °C for 72 hours. It is well known that the relative humidity of air-dried materials is estimated at temperatures of 105°C for six hours, 125°C for four hours, or 135°C for three hours, whereas the total humidity of juicy and green materials is estimated at 40-60°C for 24-72 hours. Furthermore, the estimate of total carbohydrates is supposed to be called nitrogen-free extract; it is known that it is estimated at 100- (moisture + protein + fat + fiber + ash), and non-carbohydrate fiber at 100- (moisture + protein + fat + NDF + ash). This is something that the author must correct.

Response: Thank you for your input and for providing the information. We would like to emphasize that all calculations are conducted on a dry matter basis, and we adhered to the standard protocol for drying the samples until no further reduction in weight was observed. This typically involves drying the sample until it reaches a constant weight, meaning that there is no further reduction in weight observed over successive drying periods. We were not focused on the relative humidity which is bit different concept.

3. Calculations for energy and digestible crude protein values must be combined with statistical analysis.

Response: Modified as per the suggestions.

Results:

1. Does Table 1 show the chemical composition of a fresh or silage sample?

It should be clarified in the title of the table and the text, even if it is a fresh sample, where the chemical analysis of the silage sample is, and vice versa?

Response: Thanks for pointing out, yes this is for fresh samples and changes have been made in table and within text as well.

The same trend with carbohydrates, protein fractions, and energy values in Table 2. Chemical analysis is required for five types of samples, whether silage or fresh. Why does the author review the silage analysis of selected genotype results again in the in vivo assessment? It is a salvation that was discussed during the initial evaluation. I recommend removing it from the in vivo evaluation. Thus, I recommend that the author organize the results as follows:

• Chemical composition of fresh material

• Fermentation quality and chemical composition of silage

• In vivo digestibility by sheep

Response: The table 2 also for the fresh samples and changes have been made accordingly. To review the silage analysis for the selective genotype because, our investigation, we found two lines (IG96-96 and IG96-401) exhibiting nutritional profiles comparable to maize (reported in our previous studies). Therefore, we selected only these two genotypes out of the initial five used for fresh sample analysis for further consideration. The primary objective in this study was to identify Cenchrus grass genotypes that could serve as substitutes or alternatives to maize, commonly used for silage preparation.

2. To combine silage fermentation characteristics and chemical analysis, it is advisable to present them in a single table. When writing about materials and methods, arrange them in the order of the table. It is preferable to start with the fermentation characteristics of silage and then include the chemical analysis in a table and write them in a manuscript. This approach will facilitate the interpretation of the results.

Response: We thank for your directions but regarding the fermentation quality we assessed all the five genotypes of C. ciliaris and then further we selected two genotypes (IG96-401 and IG96-96) along with maize for large scale ensiling in pits for the animal feeding. Therefore, we kept these two tables separately.

3. The results of the evaluation silage of five genotypes show that silage with IG 96-401 had the highest lactic acid, lowest ammonia concentration, and lowest pH. It was preferable to test it in vivo exclusively with maize silage. On what basis did the author select 96-96? Although the silage evaluation indicates that it is the best IG 96-50 after IG 96-401.

Response: That’s correct one. In general, all the three (IG 96-50, IG 96-401 and IG 96-96) genotypes were found suitable for the ensiling characteristics. But due to the availability of material we selected two (IG 96-401 and IG 96-96) genotypes. This work was the extension of our previous work where we screened almost 79 accessions belonging to 6 species of Cenchrus germplasm for key nutritional and silage quality traits.

(Singh, S., Singh, T., Singh, K.K., Srivastava, M.K., Das, M.M., Mahanta, S.K., Kumar, N., Katiyar, R., Ghosh, P.K. and Misra, A.K., 2023. Evaluation of global Cenchrus germplasm for key nutritional and silage quality traits. Frontiers in Nutrition, 9, p.1094763.)

4. Why does the author include corn silage in the initial assessment of five types, especially since his main goal is to use it as an alternative feed to corn silage? His genotype selection was supposed to be based on the closest assessment of corn silage as the standard.

Response: As mentioned in the response above, our main goal was to use this grass for silage without any additives. After studying for 5-6 years, we found a few genotypes with sugar content over 6-7%, making them suitable for silage. That's why we didn't initially consider maize, but we included it in the feeding trial as it's a well-known standard for comparison.

Discussion

1. In the section of the discussion, the results must be discussed as follows:

• Chemical composition of fresh material

The author should review studies that agree or disagree with the study of chemical composition of fresh sample.

Response: Suggestion is accepted, and changes have been made accordingly.

• Fermentation quality and chemical composition of silage

- As you discuss the fermentation parameters for silage, you must clarify whether each parameter in the study conforms to the specifications of good silage or not. For example, if the values of the PH of silage, lactic acid, and ammonia for the standared (Maize silage) and genotypes are recommended for the good silage and indicative of its increase or decrease than the values recommended, you must discuss, with references. Furter, auther sould discuss change chemical composition of silage (labotary ensiling).

Response: The information is incorporated into the text. The reference 54, 55 ,56. The Cenchrus genotypes are found a good tropical grasses for ensiling purpose.

- The author should discuss the reasons for his choice of the two types of genotypes for animal evaluation.

Response: Following the fermentation quality assessment, three out of the five genotypes (IG 96-50, IG 96-401, and IG 96-96) were deemed suitable for ensiling. However, due to the unavailability of material for IG 96-50, we proceeded with only these two genotypes (IG 96-401 and IG 96-96) for further chemical analysis and in vivo feeding trials. The same is incorporated in the discussion.

• In vivo digestibility by sheep

- It should be noted that if there are results that do not have an impact, such as in nutrient digestion, for example, it is essential to refer to references that either agree or disagree with the study and explain the reasons behind this discrepancy.

Response: We already have reference in relation to the digestibility correlated with the nutritional and chemical composition of the feed. 

- The author cites plants in the discussion that have nothing to do with the plant under study (underlines 459, 460, 464, and 465); however, he must also cite references to the plant under study.

Response: The unrelate content is removed, and text is modified.

Reviewer #2: 

Line 96-97. Where all the samples harvested in the same day or in different days?

Response: All the samples were collected on the same day and they were processed together and then stored at 4°C for further chemical and other studies.

Line 109. Please explain why the authors used these two genotypes (IG96-96 and IG96-401) for the sheep study.

Response: Our primary objective in this study was to identify Cenchrus grass genotype that could serve as substitutes or alternatives to maize, commonly used for silage preparation. Through our investigation, we discovered two lines (IG96-96 and IG96-401) exhibiting nutritional profiles comparable to maize. Therefore, we selected only these two lines out of the initial five for further consideration."

Line 229. Please be more specific what kind of experimental design the authors used for both green (fresh) and silage samples, and for the in vivo study.

Response: In our study, we employed a Completely Randomized Design for both the green (fresh) and silage samples, ensuring unbiased treatment allocation. For the in vivo animal study, we utilized a Randomized Block Design to enhance precision by accounting for variation among animals. The same is incorporated into the text.

Line 265 Table 2. PB1 of genotype IG96-50= 1310? I think is a typing error.

Response: Thanks for notifying yes it was typo error, now it is rectified.

Line 310 Table 6. Please report daily weight gain, if possible, this data will help the readers to visualize the effect of feeding these type of genotypes

Response: Thank you for the notification. Our primary objective was to assess the feeding value in respect of intake, nutrients digestibility and rumen fermentation. We did not specifically focus on studying animal production purposes, which is why we did not calculate body weight parameters.

---

## [Decision Letter · Decision Letter 1]

8 May 2024

PONE-D-24-07834R1Assessing genotypes of buffel grass (Cenchrus ciliaris) as an alternative to maize silage for sheep nutritionPLOS ONE

Dear Dr. Koli,

Thank you for submitting your manuscript to PLOS ONE. After careful consideration, we feel that it has merit but does not fully meet PLOS ONE’s publication criteria as it currently stands. Therefore, we invite you to submit a revised version of the manuscript that addresses the points raised during the review process.

We look forward to receiving your revised manuscript.

Kind regards,

Adham A. Al-Sagheer

Academic Editor

PLOS ONE

Journal Requirements:

**Additional Editor Comments:**

The manuscript needs thorough English language review.

Reviewers' comments:

Reviewer's Responses to Questions

**Comments to the Author**

1. If the authors have adequately addressed your comments raised in a previous round of review and you feel that this manuscript is now acceptable for publication, you may indicate that here to bypass the “Comments to the Author” section, enter your conflict of interest statement in the “Confidential to Editor” section, and submit your "Accept" recommendation.

Reviewer #1: All comments have been addressed

Reviewer #2: All comments have been addressed

2. Is the manuscript technically sound, and do the data support the conclusions?

Reviewer #1: Yes

Reviewer #2: Yes

3. Has the statistical analysis been performed appropriately and rigorously? 

Reviewer #1: (No Response)

Reviewer #2: Yes

4. Have the authors made all data underlying the findings in their manuscript fully available?

Reviewer #1: Yes

Reviewer #2: Yes

5. Is the manuscript presented in an intelligible fashion and written in standard English?

Reviewer #1: Yes

Reviewer #2: No

6. Review Comments to the Author

**Reviewer #1: **

After reviewing the paper "Assessing genotypes of buffel grass (Cenchrus ciliaris) as an alternative to maize silage for sheep nutrition," the author followed all existing comments with one minor change: the paragraph for Rumen fermentations and microbial counts must be moved from line 165 to 170, after line 210.

**Reviewer #2: **

Line 114 rephrase the sentence… mixture samples were analyzed as described (14)

Line 129 correct the sentence…Total carbohydrate (g/kg of dry matter) of green (fresh) and silage samples were determined

Lines 167-168 correct the sentence… entodiniomorphs were identified by the method of Ogimoto and Imai (25)

Line 222- 224: correct the reference style…. Fonnesbeck et al. (1984); Khalil et al. (1986).

Table 1. revise the hemicellulose value… 2158

Line 360. Correct English…An another

7. PLOS authors have the option to publish the peer review history of their article (what does this mean?). If published, this will include your full peer review and any attached files.

Reviewer #1: **Yes: **

Reviewer #2: No

---

## [Author Response · Author response to Decision Letter 1]

9 May 2024

Response to Reviewers

Reviewer# 1

After reviewing the paper "Assessing genotypes of buffel grass (Cenchrus ciliaris) as an alternative to maize silage for sheep nutrition," the author followed all existing comments with one minor change: the paragraph for Rumen fermentations and microbial counts must be moved from line 165 to 170, after line 210.

Response: Changes are accepted and modified into the manuscript.

Reviewer #2: 

Line 114 rephrase the sentence… mixture samples were analyzed as described (14)

Response: Sentence is rephrased as per the suggestion.

Line 129 correct the sentence…Total carbohydrate (g/kg of dry matter) of green (fresh) and silage samples were determined

Response: Sentence is corrected and rephrased.

Lines 167-168 correct the sentence… entodiniomorphs were identified by the method of Ogimoto and Imai (25)

Response: Sentence is corrected.

Line 222- 224: correct the reference style…. Fonnesbeck et al. (1984); Khalil et al. (1986).

Response: the reference style corrected for both.

Table 1. revise the hemicellulose value… 2158

Response: Oh! thanks for notifying. that was typo error, now rectified.

Line 360. Correct English…An another

Response: Checked for English language.

---

## [Editor Report · Decision Letter 2]

10 May 2024

Assessing genotypes of buffel grass (Cenchrus ciliaris) as an alternative to maize silage for sheep nutrition

PONE-D-24-07834R2

Dear Dr. Koli,

We’re pleased to inform you that your manuscript has been judged scientifically suitable for publication and will be formally accepted for publication once it meets all outstanding technical requirements.

Kind regards,

Adham A. Al-Sagheer

Academic Editor

PLOS ONE
---

## [Editor Report · Acceptance letter]

14 May 2024

PONE-D-24-07834R2 

PLOS ONE

Dear Dr. Koli, 

I'm pleased to inform you that your manuscript has been deemed suitable for publication in PLOS ONE. Congratulations! Your manuscript is now being handed over to our production team.

Kind regards, 

on behalf of

Dr. Adham A. Al-Sagheer 

Academic Editor

PLOS ONE